# Programmed folding into *spiro*-multicyclic polymer topologies from linear and star-shaped chains

Yoshinobu Mato[1], Kohei Honda[1], Brian J. Ree[1], Kenji Tajima [2], Takuya Yamamoto [2], Tetsuo Deguchi[3], Takuya Isono [2✉] & Toshifumi Satoh [2✉]

The development of precise folding techniques for synthetic polymer chains that replicate the unique structures and functions of biopolymers has long been a key challenge. In particular, *spiro*-type (i.e., 8-, trefoil-, and quatrefoil-shaped) polymer topologies remain challenging due to their inherent structural complexity. Herein, we establish a folding strategy to produce *spiro*-type multicyclic polymers via intramolecular ring-opening metathesis oligomerization of the norbornenyl groups attached at predetermined positions along a synthetic polymer precursor. This strategy provides easy access to the desired *spiro*-type topological polymers with a controllable number of ring units and molecular weight while retaining narrow dispersity ($Đ < 1.1$). This effective strategy marks an advancement in the development of functionalized materials composed of specific three-dimensional nanostructures.

[1] Graduate School of Chemical Sciences and Engineering, Hokkaido University, Sapporo 060-8628, Japan. [2] Division of Applied Chemistry, Faculty of Engineering, Hokkaido University, Sapporo 060-8628, Japan. [3] Department of Physics, Faculty of Core Research, Ochanomizu University, Ohtsuka 2-1-1, Bunkyo-ku, Tokyo 112-8610, Japan. ✉email: isono@eng.hokudai.ac.jp; satoh@eng.hokudai.ac.jp

Precise folding of a biopolymer chain is an essential process to attain sophisticated higher-ordered structures, such as DNA packing and three-dimensional (3D) protein structures, which is responsible for their outstanding functions in living systems[1–3]. Inspired by the folding process of biopolymers, significant efforts have been dedicated to the folding of synthetic polymers. The synthesis of topologically unique polymers from linear polymers can be regarded as mimicking biopolymer folding processes[4–6]. One successful example of this approach is the intramolecular crosslinking of linear polymers to afford single-chain nanoparticles (SCNPs) that feature a densely packed single-chain globule with a 3D nanostructure[7,8]. However, the resulting SCNP is a statistical mixture of undefined-shape chains since the intramolecularly crosslinked formations randomly occur along the main chain.

Another remarkable approach that has been demonstrated is the programmed folding of polymer chains into predetermined cyclic-type topologies[6,9,10]. The simplest case involves intramolecular coupling between the chain ends of a linear polymer to form a monocyclic polymer with unique properties attributed to the lack of chain ends[9–12]. In addition, multicyclic topological polymers (i.e., *fused-* (such as θ-shaped), *bridged-* (such as manacle-shaped), and *spiro*-multicyclic polymers (such as 8-shaped)) that consist of multiple macromolecular rings, have also been intriguing synthetic targets due to their interesting 3D structures[13,14]. Among these multicyclic topological polymers, effective construction of *spiro*-multicyclic topologies remains the most challenging due to inherently complicated architectures consisting of multiple cyclic units tethered at a single junction point.

Several synthetic strategies have been developed over the past decade to prepare *spiro*-multicyclic polymers: (i) intermolecular coupling of cyclic constituents[15,16], (ii) intermolecular cyclization of linear polymers with a multifunctionalized linker[17], and (iii) intramolecular cyclization of linear/star polymer precursors, in which functional groups are placed at the chain ends and/or chain center for multiple bond-forming reactions (such as the electrostatic self-assembly and covalent fixation (ESA-CF) protocol)[18–21]. More specifically, the intermolecular coupling of monocyclic constituents, i.e., strategy (i), can produce a series of *spiro*-multicyclic polymers with varied number of cyclic units; however, it requires the elaborated synthesis of cyclic polymers having a reactive functional group, as well as a tedious purification process to remove the excessive monocyclic reactant[15,16]. Strategy (ii) is a simple way to access an 8-shaped topology although it is limited by the formation of many possible byproducts[17]. On the other hand, strategy (iii) is advantageous in terms of suppressing by-product formation because the reaction is essentially concluded in a single polymer chain under a high-dilution condition[18–21]. However, this strategy does not exhibit synthetic versatility, such as control over the size and number of cyclic units, because it requires sophisticated preparation of highly functionalized precursors. Thus, a precise yet universal folding strategy to *spiro*-multicyclic polymers has remained elusive because increasing the number of constitutional cyclic units leads to synthetic difficulties. Owing to lack of an effective synthetic platform, only few comprehensive studies have been attempted to control the size and number of cyclic units, and thus, the structure–property relationships associated with this folded topology are not well-defined[22,23].

Our group recently reported a robust and precise strategy for constructing a variety of cage-shaped multicyclic topologies based on intramolecular consecutive cyclization (i.e., intramolecular ring-opening metathesis oligomerization (ROMO) of an *exo*-norbornenyl group attached to each terminus of star-shaped polymers), which enabled systematic synthesis and

characterization[24]. A unique feature of our strategy is that polymerizable *exo*-norbornenyl groups at the chain termini are immediately transformed for intramolecular chain-growth upon addition of excess Grubbs' 3rd catalyst (G3). We envisaged that the aforementioned challenges in *spiro*-multicyclic polymer synthesis, i.e., synthetic simplicity and versatility to allow access to a series of polymers with varied size and number of cyclic units, can be overcome by applying this novel strategy. Herein, we demonstrate the utility of intramolecular ROMO to accomplish programmed folding into *spiro*-multicyclic polymer topologies. This is the first example, to our knowledge, of the synthesis of inherently complex *spiro*-type multicyclic architectures through the intramolecular chain reaction to facilitate cyclic unit formation.

## Results and discussion

**Construction of an 8-shaped polymer.** For synthesis of an 8-shaped polymer, we initially designed a poly(ε-caprolactone) (PCL)-based linear precursor with norbornenyl groups at each ω-chain-end and chain center (**P2**; Fig. 1). To introduce the norbornenyl groups at not only the chain ends but also the chain center, a diol initiator possessing a protected hydroxyl group was employed for the polymerization. The precursor of **P2** [**P2-a**; molecular weight from [1]H nuclear magnetic resonance (NMR) $(M_{n,NMR})$ = 6200 g mol$^{-1}$, molecular weight from size-exclusion chromatography (SEC; $M_{n,SEC}$) = 9970 g mol$^{-1}$, dispersity $(Đ)$ = 1.06] was successfully synthesized by the ring-opening polymerization of ε-caprolactone using the initiator, followed by deprotection and condensation with (±)-*exo*-5-norbornene carboxylic acid.

Subsequently, the intramolecular ROMO was carried out under very dilute conditions in $CH_2Cl_2$ (final concentration = 0.02 mM) in the presence of excess Grubbs' 3rd generation catalyst ([precursor]$_0$/[G3] = 1/6) with slow addition of the polymer precursor. After removing the catalyst residue, the product was obtained in high yield (91.0%).[1]H NMR analysis of the obtained product revealed that signals arising from the *exo*-norbornenyl groups of the precursor completely disappeared, while new signals due to the oligonorbornene backbone appeared at 1.6–3.4 and 4.8–6.5 p.p.m. (Supplementary Fig. 1e). The SEC trace of the product showed a monomodal peak, and the elution peak was observed in a lower molecular weight region $(M_{n,SEC} = 7340$ g mol$^{-1}$, $Đ = 1.08)$ than that of the linear precursor (**P2-a**; $M_{n,SEC}$ = 9970 g mol$^{-1}$, $Đ = 1.06$, Fig. 2a and Supplementary Fig. 2). The dramatic shift in the SEC elution peak maximum indicates that intramolecular cyclization occurred to produce a hydrodynamically smaller polymer (i.e., **MC2-a**), rather than intermolecular side reactions. By comparing the SEC peak-top molecular weights of the precursors and cyclized polymers, the compaction parameter <G> was calculated as 0.67, which agrees with the reported value of <G> for 8-shaped polymers of comparable molecular weights (<G> = 0.67–0.81; Table 1)[16,18,19,25–27]. The matrix-assisted laser desorption/ionization-time of flight mass spectrometry (MALDI-TOF MS) spectrum of the obtained product showed two series of peaks with a regular $m/z$ interval of 114.09, corresponding to the mass of the repeating ε-caprolactone unit (Fig. 2b). The major series of peaks (denoted with •) was assigned to the expected chemical structure of **MC2-a**; for example, the peak at $m/z = 5184.84$ agrees well with the calculated mass of the desired **MC2-a** with a total degree of polymerization of 40 ([M + Na]$^+$ = 5184.12). The minor series of peaks (denoted with ■) was assigned to cyclic or tadpole-shaped PCLs (calculated [M + Na]$^+$ = 5174.11, $n = 39$) formed by reaction with two G3 molecules (Supplementary Fig. 3). The amount present was determined to be 7.9% by peak

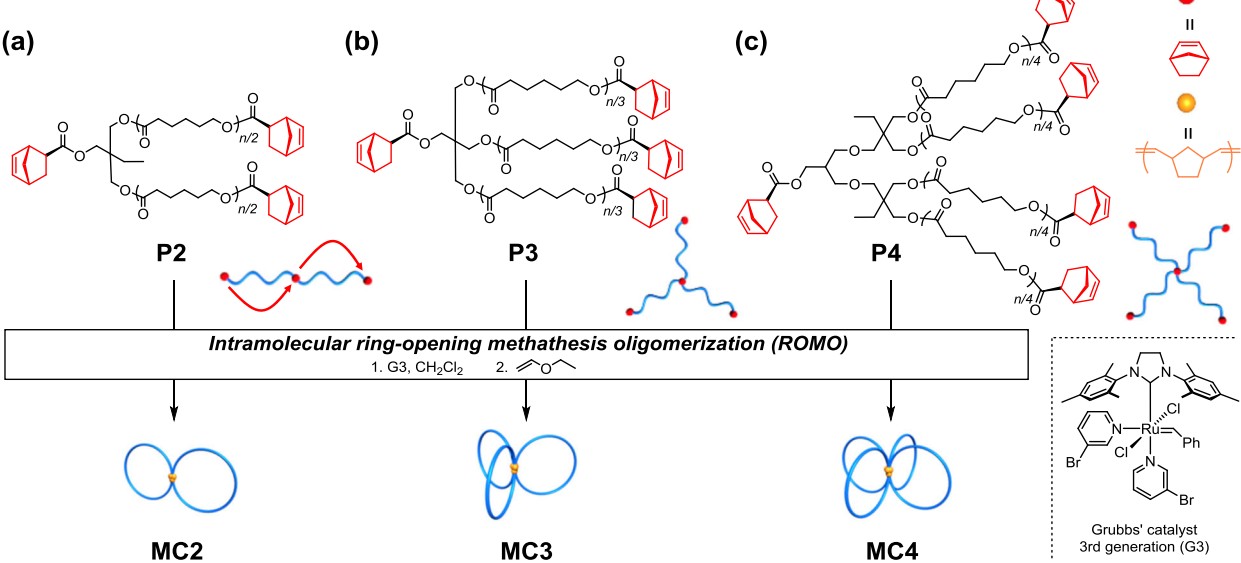

**Fig. 1 Folding strategy to *spiro*-multicyclic polymers.** Programmed folding of linear, three-armed star, and four-armed star precursors with norbornenyl groups at the chain center and each end into Fig. 8- (**MC2**; **a**), trefoil- (**MC3**; **b**), and quatrefoil-shaped PCLs (**MC4**; **c**) through intramolecular ROMO.

deconvolution. Two topological isomers of cyclic- and tadpole-forms are expected to be produced by a side reaction, with the preferred isomer expected to have the tadpole topology due to the folding process being primarily directed by the spatial distance of crosslinking points[21]. Other possible byproducts (e.g., linear PCL; calculated $[M + Na]^+ = 5278.17$, $n = 39$) were not detected. Overall, successful folding into the 8-shaped topology (**MC2-a**) was achieved by intramolecular ROMO of the linear precursor in a precise manner.

**Construction of trefoil- and quatrefoil-shaped polymers.** To further extend this strategy, intramolecular ROMO was applied to the synthesis of trefoil- (**MC3**) and quatrefoil-shaped PCLs (**MC4**), where the number of cyclic units are three and four, respectively. For the synthesis of **MC3-a** and **MC4-a**, the well-defined three- and four-armed star-shaped PCLs bearing an *exo*-norbornenyl group at the chain center and each terminus (**P3-a** and **P4-a**; $M_{n,NMR} = \sim6500$ g mol$^{-1}$) were subjected to ROMO conditions (Supplementary Figs. 4–9). $^1$H NMR analysis confirmed quantitative consumption of the *exo*-norbornenyl groups despite an increase in the number of groups in the precursors (Supplementary Figs. 5 and 8). Importantly, in both cases, the SEC elution peak clearly shifted to the lower molecular weight region while the peak shape remained monomodal with narrow dispersity ($Đ = 1.07$–$1.09$; Fig. 2c, e). The estimated $<G>$ values for **MC3-a** and **MC4-a** ($0.57$–$0.58$) were much lower than that of **MC-2a** ($0.67$), confirming the much smaller volume. Notably, each MALDI-TOF MS spectrum showed only one series of peaks assignable to the expected structures of **MC3-a** and **MC4-a** (Fig. 2d, f). The SEC and MALDI-TOF MS analyses strongly support that ROMO proceeds in an intramolecular manner without side reactions regardless of the number of cyclic units. A question therefore arises with respect to the reaction order of the norbornenes. According to a recent report by Tezuka and co-workers[21], it can be reasonably expected that the norbornenes should react with those that are closer. However, to fully understand the folding process, further investigation is necessary, which is currently ongoing in our laboratory.

**Functionalization of topological polymers.** Chain-end functionalization of topological polymers is essential to facilitate higher-order functions with a combination of diverse molecular designs. Typically, the α- and ω-chain ends of the polynorbornene backbone produced by ROMO can be readily transformed to the desired reactive groups by using functional Ru initiators and end-capping agents, respectively[28,29]. Indeed, the α-/ω-end-functionalized trefoil-shaped PCLs with hydroxyl groups were precisely synthesized while retaining narrow dispersity ($Đ = 1.12$–$1.13$, Supplementary Figs. 10-14). This allows facile access to surface-modified metal/semi-conductor nanoparticles and substrates with topological polymers, which could be used to increase colloidal stability and to prepare bioinert and superlubricating coatings[30–32].

**Size control of multicyclic polymers.** For systematic characterization of the folded polymers, size control of the cyclic unit and control over the number of cyclic units is indispensable. By simply changing the degree of polymerization of the precursor, the molecular weight of each multicyclic (8-, trefoil-, and quatrefoil-shaped) polymer was successfully controlled from ~6000 to 12,000 g mol$^{-1}$ (Table 1). Note that the suffix on the name of each polymer sample represents its molecular weight (**-a** for ~6,000 g mol$^{-1}$, **-b** for ~9,000 g mol$^{-1}$, and **-c** for ~12,000 g mol$^{-1}$).

**Applicability of intramolecular ROMO to diverse polymer species.** Furthermore, to verify the applicability of intramolecular ROMO to other polymer backbones, we applied this approach to the synthesis of *spiro*-multicyclic poly(L-lactide) (PLLA) and poly (2-ethylhexyl glycidyl ether) (PEHGE) (Supplementary Figs. 15-21 and Supplementary Table 4). Specifically, trefoil-shaped PLLA and PEHGE were synthesized by the ring-opening polymerization of the corresponding monomers with **I3** as the initiator using 1,8-diazabicyclo[5.4.0]-7-undecene (DBU) and $t$-Bu-P$_4$ catalysts[33,34], respectively, followed by the deprotection reaction, installation of norbornene groups, and ROMO under the optimized conditions. The targeted folded structures were confirmed in both synthesized products through the comprehensive characterization by $^1$H NMR, SEC, and MALDI-TOF MS, which

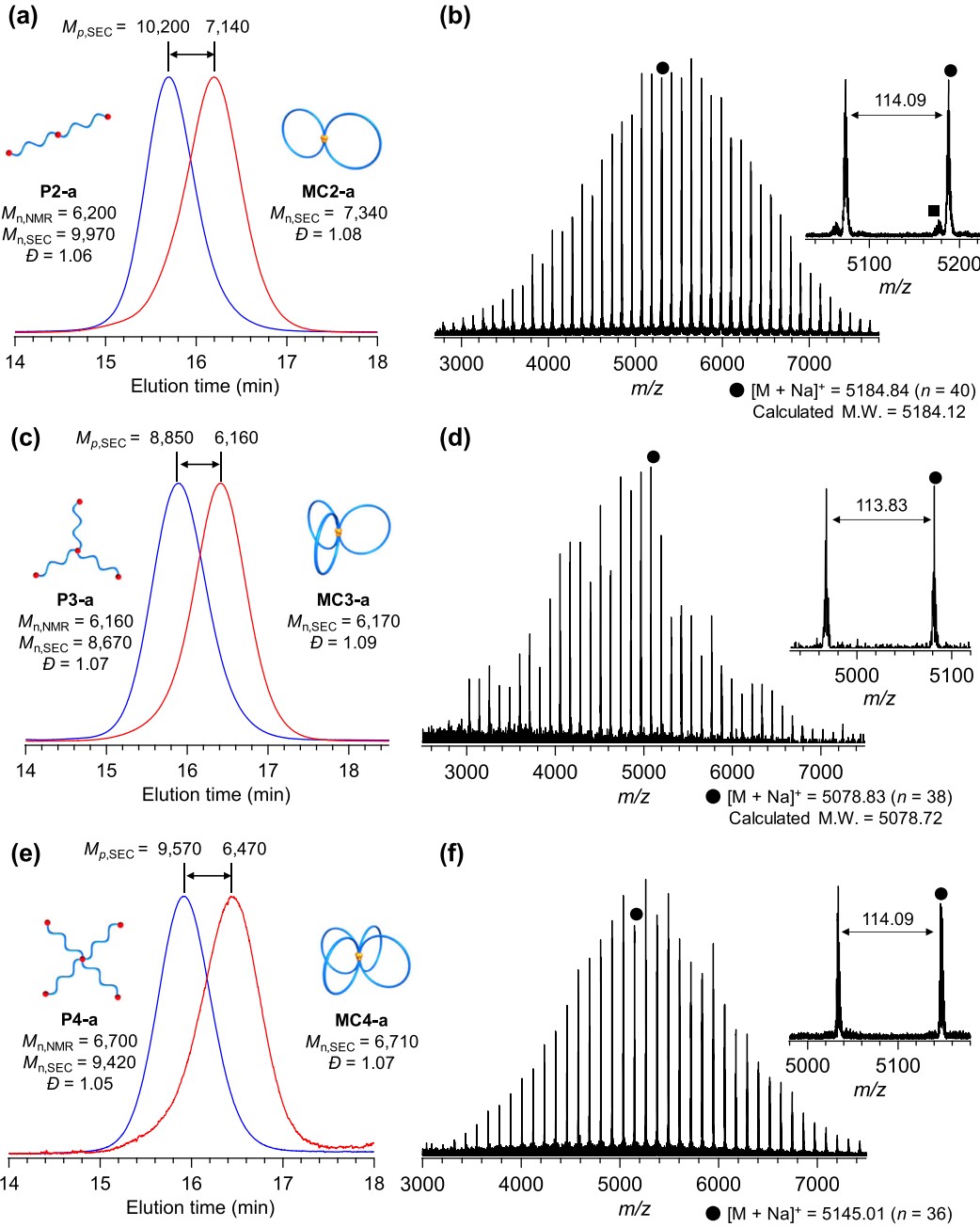

**Fig. 2 Structural characterization of the folded *spiro*-multicyclic polymers. a, c, e** SEC traces of the precursors (**P2-a**, **P3-a**, and **P4-a**; blue curves) and the obtained *spiro*-multicyclic polymers (**MC2-a**, **MC3-a**, and **MC4-a**; red curves) (RI detection; PS standards; eluent, THF). **b, d, f** MALDI-TOF MS spectra of **MC2-a**, **MC3-a**, and **MC4-a** (reflector mode; *n* denotes number of monomer units).

suggested that the presented method is applicable for the synthesis of a broad range of polymer species.

**Programmed folding into a predetermined topology via intramolecular ROMO under different conditions**. Here, it is important to note that the present folding strategy affords a polymer with a predetermined topology, as opposed to the synthesis of SCNPs, in which the size and conformation of the resulting product are considerably affected by the solvent quality[35]. To provide a proof-of-concept of our programmed folding, we performed intramolecular ROMO under different conditions that could affect the polymer chain dimension during the reaction (Fig. 3). For example, to attain the complicated multicyclic folding of **MC4-a**, different solvents and/or elevated temperatures

were used to determine whether the folded structure of the resulting product is affected. Notably, the products obtained from the intramolecular ROMO of **P4-a** in $CH_2Cl_2$/*n*-hexane showed exactly the same peak-top as **MC4-a** in the SEC, even in the case of *n*-hexane-rich media with up to 60% *n*-hexane (Fig. 3c–e), indicating successful folding into the same architecture. By switching the solvent to toluene, successful formation of **MC4-a** was also observed upon heating (Fig. 3f, g). Thus, these results suggest that the precursor can be unambiguously folded into the predetermined topology.

**Systematic investigation of structure–property relationships**. Hydrodynamic diameter ($D_h$) and intrinsic viscosity ([$\eta$]), which are correlated to the polymer chain dimensions, are good measures

**Table 1 Molecular characterization of *spiro*-multicyclic PCLs prepared by intramolecular ROMO.**

| Precursor | $M_{n,NMR(Pre)}$[a] (g mol$^{-1}$) | Multicyclic topology | $M_{n,SEC}$[b] (g mol$^{-1}$) | $Đ$[b] | $M_{p,MC}$[c] (g mol$^{-1}$) | $<G>$[d] | $D_h$[e] (nm) | $[\eta]$[e] (mL g$^{-1}$) | $T_m$[f] (°C) | $X_{WAXD}$[g] (%) |
|---|---|---|---|---|---|---|---|---|---|---|
| P2-a | 6200 | MC2-a | 7340 | 1.08 | 7140 | 0.67 | 4.2 | 9.5 | 53.8 | 41.8 |
| P2-b | 8900 | MC2-b | 10,600 | 1.07 | 10,000 | 0.65 | 5.0 | 12.3 | 56.1 | 44.5 |
| P2-c | 11,000 | MC2-c | 13,200 | 1.08 | 11,900 | 0.63 | 5.6 | 14.9 | 57.7 | 50.6 |
| P3-a | 6160 | MC3-a | 6170 | 1.09 | 6160 | 0.58 | 3.8 | 6.9 | 42.6 | 31.7 |
| P3-b | 9700 | MC3-b | 9790 | 1.08 | 9010 | 0.54 | 4.8 | 11.3 | 52.8 | 42.8 |
| P3-c | 12,400 | MC3-c | 13,200 | 1.08 | 12,300 | 0.58 | 5.6 | 13.2 | 54.4 | 45.4 |
| P4-a | 6620 | MC4-a | 6710 | 1.07 | 6470 | 0.57 | 4.0 | 6.9 | 29.3 | 23.6 |
| P4-b | 9220 | MC4-b | 9640 | 1.06 | 9140 | 0.57 | 4.4 | 8.6 | 47.7 | 39.3 |
| P4-c | 11,500 | MC4-c | 11,000 | 1.06 | 10,500 | 0.53 | 5.0 | 9.8 | 50.8 | 44.4 |

[a]The absolute molecular weight of the precursor ($M_{n,NMR(Pre)}$) estimated by $^1$H NMR in CDCl$_3$ (400 MHz).
[b]Determined by SEC in tetrahydrofuran (THF) using polystyrene (PS) standards (RI detection).
[c]Peak-top molecular weight of the multicyclic polymer ($M_{p,MC}$) estimated by SEC in THF using PS standards (refractive index (RI) detection).
[d]$<G> = M_{p,MC}/M_{n,NMR(Pre)}$, where $M_{p,MC}$ as a PS equivalent was converted into that of the PCL using a conversion coefficient of 0.58[42].
[e]Determined by SEC equipped with a viscometer in THF.
[f]Determined by differential scanning calorimetry (DSC).
[g]Determined by wide-angle X-ray diffraction (WAXD) through peak deconvolution.

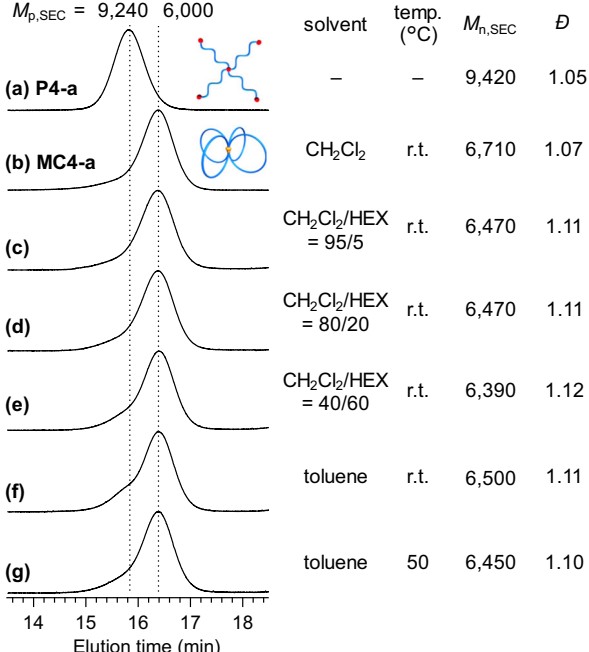

**Fig. 3 Polymer folding under various conditions.** Programmed folding from **P4-a** (**a**) into **MC4-a** (**b**) via intramolecular ROMO of the precursor with altered chain dimension induced using poor solvents (CH$_2$Cl$_2$/$n$-hexane (HEX); **c**–**e**) and variation of temperature in toluene (**f**, **g**).

to understand the folded polymer structure. To get information about the polymer chain dimensions, the obtained *spiro*-multicyclic PCLs and their related linear and cyclic counterparts were subjected to online SEC measurement combined with light scattering, viscosity, and reflective index detectors (SEC-MALS-Visco) in THF. Note that the previously synthesized linear and monocyclic counterparts with comparable molecular weights were subjected to these analyses for comparison[24]. As shown in Supplementary Fig. 22, both the $D_h$ and $[\eta]$ values decreased in the order of linear > cyclic > 8-shaped (**MC2**) > trefoil-shaped (**MC3**) > quatrefoil-shaped (**MC4**) when comparing the topologically different polymers with comparable molecular weights. This demonstrates that the polymer chain dimensions decrease with increasing number of cyclic units when the total molecular weight remains the same. Such a trend

matches very well with the theoretically predicted one; the radius of gyration decreased in the same order as reported by Deguchi and colleagues[36]. Interestingly, the aforementioned $[\eta]$ values of *spiro*-multicyclic polymers were smaller than those of the topologically related cage-shaped PCLs (7.4–27.7 mg mL$^{-1}$ for cyclic, three-arm cage-shaped, and four-arm cage-shaped)[20]. The *spiro*-multicyclic polymer is assumed to be a topological analog of the cage-shaped polymer, whereby an additional constraint at the focal point of the *spiro*-multicyclic polymer exists, further decreasing the chain dimension.

While the solution properties are important for understanding the folded structure, understanding the bulk state properties also provides new insight into topological effects. Initially, we performed thermogravimetric analysis (TGA) for the *spiro*-multicyclic PCLs with $M_{n,NMR}$ of ~6000 g mol$^{-1}$ to examine the structure–thermal degradation relationship. The TGA results revealed a negligible difference in the degradation temperature ($T_d$) for 10% weight loss among the *spiro*-type multicyclic PCLs and the corresponding linear and cyclic counterparts (384–390 °C), which suggested that the polymer topology has no significant impact on the thermal degradation of PCL (See Supplementary Fig. 30 and Supplementary Table 5). A similar conclusion was drawn in a previous report by Grayson, in which a cyclic PCL and its precursor were compared[37]. Among a number of cyclic polymers reported thus far, cyclic PCLs have been a target of investigation to understand the effect of cyclic topology on polymer crystallization behavior (e.g., melting point, crystallite size, crystallinity, and crystallization kinetics). For example, cyclic PCLs are known to possess a higher melting point and crystallinity than their linear counterparts, albeit the reason remains unclear[38–40]. With a series of well-defined *spiro*-type PCLs in hand, we thus investigated the melting temperature ($T_m$) and crystallinity ($X_{WAXD}$) using differential scanning calorimetry (DSC) and wide-angle X-ray diffraction (WAXD), respectively. Note that previously synthesized linear and monocyclic PCLs with comparable molecular weight were subjected to these analyses for comparison[24]. Figure 4 shows the $T_m$ and $X_{WAXD}$ for each architecture, which seem to correlate with both the number of cyclic units and total molecular weight. The important finding here is that 8-shaped PCLs exhibit higher melting points and crystallinity than cyclic PCLs (Fig. 4a, b). More specifically, **MC2-a** with molecular weight of ~6000 g mol$^{-1}$ exhibited dramatically enhanced $T_m$ (53.8 °C) and $X_{WAXD}$ (41.8%) compared with its linear and cyclic counterparts ($T_m$ = 43.2–51.1 °C and $X_{WAXD}$ = 35.2–40.3%). This can be explained by the

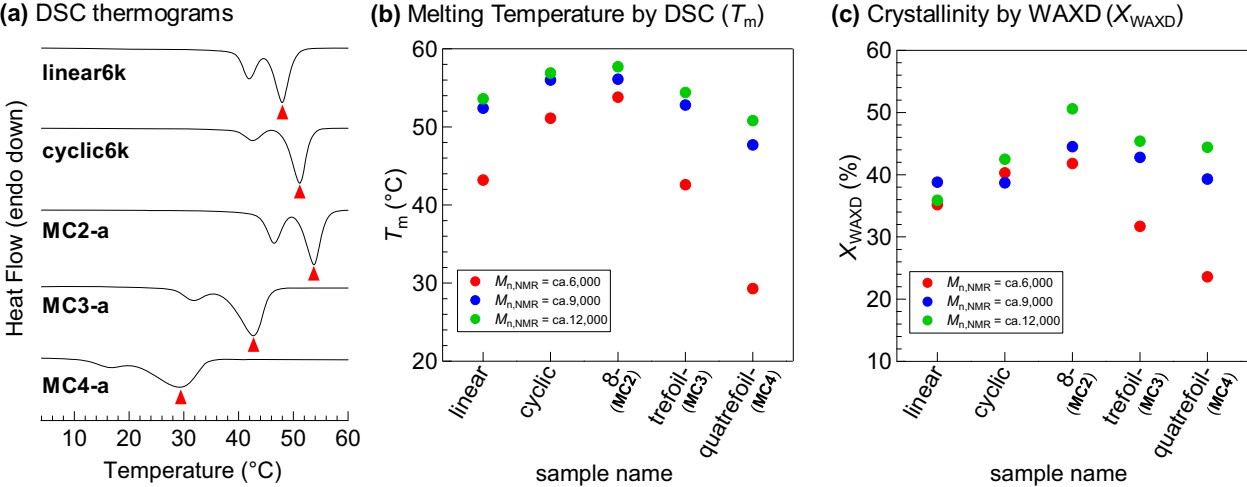

**Fig. 4 Crystallization behavior of folded *spiro*-multicyclic PCLs. a** DSC thermograms during the 2nd heating of the multicyclic polymers (**MC2-a**, **MC3-a**, and **MC4-a**) and their counterparts with molecular weights of ~6000 g mol$^{-1}$. The melting temperature ($T_m$) was determined as the peak-top of the transition marked with a triangle. **b, c** Plots of $T_m$ and $X_{WAXD}$ for the linear, monocyclic, and multicyclic polymers (**MC2**s, **MC3**s, and **MC4**s) with different molecular weights.

topological confinement of the non-crystallizable segments (Supplementary Fig. 23). The non-crystallizable segments, such as the initiator moiety and oligonorbornene backbone in the cyclic PCL have mobility, which suppresses the ordered packing of the PCL chains, resulting in lesser crystallization ability. However, the initiator moiety and oligonorbornene backbone in the 8-shaped PCL are all constrained at the focal point, which successfully reduces the random placing of non-crystallizable segments, thus resulting in better crystallization ability. Further increase in the number of cyclic units in **MC3** and **MC4** apparently lowers the $T_m$ and crystallinity. This can be attributed to suppressed molecular mobility and chain-packing ability due to strong constraints induced with increasing the number of arms. Small-angle X-ray scattering (SAXS) further supports this unique trend in lamellae thickness that appears to be correlated with $T_m$, while the long period (~12 nm) exhibits almost no change (Supplementary Note 1 and Supplementary Fig. 24), which is consistent with a previous report[41]. Compared to its counterparts, **MC2-b** showed the longest lamellae thickness of 5.1 nm, which is approximately 25% of the theoretical extended length of 20 nm for each of its arms (see the SI for details). This suggests that the *spiro*-type topological chain is likely further folded in the crystalline domain multiple times.

In summary, we demonstrated the programmed folding of linear and star-shaped polymer precursors into *spiro*-multicyclic polymers via intramolecular ROMO of the norbornenyl groups at predetermined positions. With the present strategy, diverse *spiro*-type multicyclic topologies with different amounts of cyclic units and total molecular weight were successfully constructed, demonstrating the versatility of this strategy as an effective means to synthesize topological polymers. Remarkably, this comprehensive study on the structure–property relationships of the folded PCLs revealed enhanced crystallization ability in the 8-shaped topology. Polymer folding into a *spiro*-type topology plays a crucial role in rendering higher-ordered functions to biomacromolecules. Furthermore, the results of this study can be applied to other polymer species for the development of bioinspired materials with specific 3D nanostructures.

## Methods
**Materials and instruments**. See Supplementary Methods.

**General synthetic procedures**. See Supplementary Methods.

**Characterization**. See Supplementary Figs. 22–30, Supplementary Tables 1–5, and Supplementary Note 1.

## Data availability
All data are available from the authors upon reasonable request.

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

## Acknowledgements

We gratefully acknowledge financial support from the MEXT Grant-in-Aid for Challenging Exploratory Research (16K14000 and 19K22209), Grant-in-Aid for Scientific Research (B) (19H02769), Grant-in-Aid for Scientific Research on Innovative Areas "Hybrid Catalysis" (18H04639 and 20H04798), and JST CREST (Grant Number JPMJCR19T4). This work was, in part, performed under the approval of the Photon Factory Program Advisory Committee (Proposal No. 2017G589 and 2019G579).

## Author contributions

T.I and T.S. designed the experiment. Y.M. wrote the manuscript. Y.M. and K.H. synthesized and characterized the polymers and analyzed all the data. B.J.R., T.Y., T.D, and K.T. contributed to discussion. K.T., T.I., and T.S. supervised the work. The manuscript was written through the contributions of all authors. All authors have given their approval to the final version of the manuscript.

## Competing interests

The authors declare no competing interests.
