## [Peer Review file · Communications Chemistry]

Reviewers' comments:

Reviewer #1 (Remarks to the Author):

Satoh et al has presented in this manuscript a synthetic approach toward spiro-type multicyclic polymers based on intramolecular ring-opening metathesis oligomerization of norbornenyl groups attached at both chain ends and center (initiating site) of the linear and star-shaped precursors. It is a nice extension of the method previously established by the same authors for synthesis of macromolecular cages. The study is thorough, the structure obtained are new and interesting, and the conclusions are generally well supported by the data. I recommend publishing this work after a minor revision. Specific comments are given below.

1. For the initial feed ratio of [precursor]₀/[G3], why is it kept constant at 1/6?
2. It would be more interesting to know something about the structure-property relationship in terms of the (bio)degradation of these PCLs.
3. Since this is not the first time this method is reported, can the authors provide some evidence that it can be really "applied to other polymer species". What effect could the inherent properties (e.g. rigidity) of polymer chains have on the intramolecular multi-cyclization?

Reviewer #2 (Remarks to the Author):

This manuscript demonstrated the synthesis of spiro-multicyclic polymers by programmed folding of linear and star-shaped polymer precursors via intramolecular ROMO. Overall, this is an interesting research work in cyclic polymer field. The experimental and characterization work was well done. I recommend publication after suitable revisions.

1. One main concern is the authors did not explain the novelty or advantages of the synthetic method of this paper clearly, as there are many examples for constructing spiro-multicyclic polymers, such as *Macromol. Rapid Commun.* 2008, 29, 1672–1678, *Macromolecules* 2014, 47, 2853–2863, *Macromolecules* 2017, 50, 1463–1472, *ACS Macro Lett.* 2017, 6, 1036–1041, *Polym. Chem.* 2019, 10, 3895-3901.
2. The units of molecular weights need to be added.
3. Synthesis of linear and cyclic PCLs is not presented or mentioned in the manuscript.
4. For ref 27, "Macromolecules 28" should be moved after the title.
5. M_n , SEC values of P4-a and MC4-a in Fig. 2 (9420 and 6710) are inconsistent with those in Fig. 3 (9360 and 6490), and \bar{D} of MC4-a in Fig. 3 is 1.11, however, in Table 1, the value is 1.07.

In the responses, the page and line numbers reflect those in the originally submitted version.

Reviewer #1

1. For the initial feed ratio of [precursor]₀/[G3], why is it kept constant at 1/6?

Response: As we reported in our previous paper (*Chem. Sci.* **2019**, *10*, 440-446), the ratio of [precursor]₀/[G3] directly affects the cyclization efficiency; a low [precursor]₀/[G3] ratio leads to the formation of undesired oligomers through intermolecular propagation. The same side reaction is highly expected for the current system. Indeed, we found oligomer formation in the cyclization of the **P2** precursor by applying low [precursor]₀/[G3] ratios even under the highly diluted condition. Thus, we have employed the optimized [precursor]₀/[G3] ratio of 1/6 to achieve precise synthesis.

2. It would be more interesting to know something about the structure-property relationship in terms of the (bio)degradation of these PCLs.

Response: According to the reviewer's comment, we have additionally performed thermogravimetric analysis (TGA) on the *spiro*-multicyclic PCLs to examine the structure-thermal degradation relationship. To this end, a series of PCL samples with total MW of 6,000 g mol⁻¹ was selected and subjected to the analysis (see Table R1 for summary). Because the TGA results (see Supplementary Fig. S29 and Supplementary Table S5) revealed a negligible difference in the degradation temperature (*T*_d) for 10% weight loss over the range of 384–390 °C, we concluded that the polymer topology has no significant impact on the thermal degradation behavior of PCL. A similar conclusion was drawn in a previous report by Grayson group, in which a cyclic PCL and its precursor were compared (*Macromolecules* **42**, 6406–6413 (2009)).

Figure S29. TGA result of *spiro*-multicyclic PCLs and their linear and cyclic counterparts obtained under Ar atmosphere (total MW; ~6000 g mol⁻¹, heating rate; 10 °C min⁻¹).

Table S5. Thermal degradation properties of *spiro*-multicyclic polymers and their linear and cyclic

counterparts

Sample	$M_{n,NMR}$ (precursor) (g mol ⁻¹)	T_d (°C) ^a
linear _{6k}	6,540	384
cyclic _{6k}	6,540	387
MC2-a	6,200	389
MC3-a	6,160	388
MC4-a	6,620	390

^a 10% degradation temperature (T_d) was determined by TGA.

Accordingly, we have added following sentences at lines 10–16 of page 12 (or at lines 11–18 of page 14 in the revised manuscript) to briefly comment on the TGA results:

“Initially, we performed thermogravimetric analysis (TGA) for the *spiro*-multicyclic PCLs with $M_{n,NMR}$ of $\sim 6,000$ g mol⁻¹ to examine the structure–thermal degradation relationship. The TGA results revealed a negligible difference in the degradation temperature (T_d) for 10% weight loss among the *spiro*-type multicyclic PCLs and the corresponding linear and cyclic counterparts (384–390 °C), which suggested that the polymer topology has no significant impact on the thermal degradation of PCL (See Supplementary Fig. S29 and Supplementary Table S5). A similar conclusion was drawn in a previous report by Grayson, in which a cyclic PCL and its precursor were compared.³⁸”

As the reviewer mentioned, the biodegradation behavior of the multicyclic polymers is also of great interest. However, demonstrating this behavior is another big challenge in the field of polymer science. To understand the relationship between biodegradability and polymer topology, considerable research effort and long-term experiments are needed. In future studies, we would like to investigate this relationship and would like to report the results in a separate paper.

3. Since this is not the first time this method is reported, can be authors provide some evidence that it can be really “applied to other polymer species”. What effect could the inherent properties (e.g. rigidity) of polymer chains have on the intramolecular multi-cyclization?

Response: We thank the reviewer’s valuable comment very much. To demonstrate the applicability of the proposed method over a wide range of polymer species, we additionally performed the syntheses of trefoil-shaped poly(L-lactide) (PLLA; as polylactide) and poly(2-ethylhexyl glycidyl ether) (PEHGE; as polyether) using the presented methodology. Careful structural characterization revealed the successful synthesis of the targeted trefoil-shaped polymers with different backbones. Accordingly, we added a new paragraph at lines 10–23 in page 10 (or at lines 11–19 of page 11 in the revised manuscript) as follows:

“Furthermore, to verify the applicability of intramolecular ROMO to other polymer backbones, we applied this approach to the synthesis of *spiro*-multicyclic poly(L-lactide) (PLLA) and poly(2-ethylhexyl glycidyl ether) (PEHGE) (the synthetic details can be found in

Supplementary Information; Supplementary Scheme S17, Supplementary Figs. S15–20, and Supplementary Table S4). Specifically, trefoil-shaped PLLA and PEHGE were synthesized by the ring-opening polymerization of the corresponding monomers with **I3** as the initiator using 1,8-diazabicyclo[5.4.0]-7-undecene (DBU) and *t*-Bu-P₄ catalysts,^{33,34} respectively, followed by the deprotection reaction, installation of norbornene groups, and ROMO under the optimized conditions. The targeted folded structures were confirmed in both synthesized products through the comprehensive characterization by ¹H NMR, SEC, and MALDI-TOF MS, which suggested that the presented method is applicable for the synthesis of a broad range of polymer species.”

We think that the inherent properties of polymer chains, such as backbone rigidity and side chain bulkiness, would affect the cyclization efficiency. However, to demonstrate this effect, considerable research effort and long-term experiments are needed. We are currently trying to apply the presented method to various functional polymers, through which we can get more information about the effect of the inherent polymer properties on the polymer folding process. In future studies, we would like to investigate this aspect and would like to report the results in a separate paper.

Reviewer #2

1. One main concern is the authors did not explain the novelty or advantages of the synthetic method of this paper clearly, as there are many examples for constructing spiro-multicyclic polymers, such as *Macromol. Rapid Commun.* 2008, 29, 1672–1678, *Macromolecules* 2014, 47, 2853–2863, *Macromolecules* 2017, 50, 1463–1472, *ACS Macro Lett.* 2017, 6, 1036–1041, *Polym. Chem.* 2019, 10, 3895-3901.

Response: We thank the reviewer very much for the valuable and kind comment. The most important novelty of the presented synthetic method is the use of the intramolecular chain reaction for cyclic unit formation, resulting in a series of inherently complex *spiro*-type multicyclic architectures. This method is highly advantageous over the previously reported synthetic methods in terms of the simplicity and versatility. For example, the intermolecular coupling of monocyclic constituents can produce a series of *spiro*-multicyclic polymers with varied number of cyclic units, although it requires the elaborated synthesis of cyclic polymers having a reactive functional group as well as a tedious purification process to remove the excessive monocyclic reactant.^{15,16} The intermolecular coupling of a linear polymer with a tetrafunctional crosslinker is a simple way to access an 8-shaped polymer; however, it is limited by the formation of many possible byproducts.¹⁷ Highly pure multicyclic polymers can be synthesized via the intramolecular cyclization approach under the high dilution condition.^{18–21} However, in this case, a sophisticated synthesis of the highly functionalized precursor is needed, implying its limitation of synthetic versatility. Thus, each known synthetic approach has serious disadvantages. Such synthetic difficulties have primarily limited the access to a series of *spiro*-type multicyclic polymers with varied ring size and number of rings. On the other hand, our approach overcame the disadvantages such as the elaborated preparation of acyclic precursors, byproduct formation, and tedious purification, allowing systematic synthesis and investigation of the topology–physical property relationship of the *spiro*-type multicyclic polymers.

To further clarify the merits of this study, comparison between the presented method and previously reported synthetic methods for *spiro*-multicyclic polymers is included at lines 8–19 and 19–21 in page 4 and lines 1–3 in page 5 (or at lines 8–17 of page 4 and lines 9–14 of page 5 in the revised manuscript) as follows:

“More specifically, the intermolecular coupling of monocyclic constituents, i.e., strategy (i), can produce a series of *spiro*-multicyclic polymers with varied number of cyclic units; however, it requires the elaborated synthesis of cyclic polymers having a reactive functional group as well as a tedious purification process to remove the excessive monocyclic reactant.^{15,16} Strategy (ii) is a simple way to access an 8-shaped topology although it is limited by the formation of many possible byproducts.¹⁷ On the other hand, strategy (iii) is advantageous in terms of suppressing by-product formation because the reaction is essentially concluded in a single polymer chain under a high dilution condition.^{18–21} However, this strategy does not exhibit synthetic versatility, such as control over the size and number of cyclic units, because it requires sophisticated preparation of highly functionalized precursors. Thus, a precise yet universal folding strategy to *spiro*-multicyclic polymers has remained elusive because increasing the number of constitutional cyclic units leads to synthetic difficulties.”

“We envisaged that the aforementioned challenges in *spiro*-multicyclic polymer synthesis, i.e., synthetic simplicity and versatility to allow access to a series of polymers with varied size and number of cyclic units, can be overcome by applying this novel strategy. Herein, we demonstrate the utility of intramolecular ROMO to accomplish programmed folding into *spiro*-multicyclic polymer topologies. This is the first example of the synthesis of inherently complex *spiro*-type multicyclic architectures through the intramolecular chain reaction to facilitate cyclic unit formation.”

2. The units of molecular weights need to be added.

Response: According to the reviewer’s comment, the unit of molecular weights “g mol⁻¹” has been added at all instances in the main text and SI as follows:

For example, at line 7–12 in main text page 5: “The precursor of **P2** [**P2-a**; molecular weight from ¹H nuclear magnetic resonance (NMR) ($M_{n,NMR}$) = 6,200 g mol⁻¹, molecular weight from size-exclusion chromatography (SEC; $M_{n,SEC}$) = 9,970 g mol⁻¹, dispersity (D) = 1.06] was successfully synthesized by the ring-opening polymerization of ϵ -caprolactone using the initiator, followed by deprotection and condensation with (\pm)-*exo*-5-norbornene carboxylic acid (Supplementary Schemes S1–4).”

3. Synthesis of linear and cyclic PCLs is not presented or mentioned in the manuscript.

Response: We thank the reviewer very much for the comment. The previously synthesized

linear (polymer precursor of cyclic PCL) and cyclic PCLs with comparable molecular weights were used for the analyses. According to the reviewer's comment, the following sample details have been added at lines 7–8 in page 11 (or at lines 16–17 of page 13 in the revised manuscript):

“To get information about the polymer chain dimensions, the obtained *spiro*-multicyclic PCLs and their related linear and cyclic counterparts were subjected to online SEC measurement combined with light scattering, viscosity, and reflective index detectors (SEC-MALS-Visco) in THF. Note that the previously synthesized linear and monocyclic counterparts with comparable molecular weights were subjected to these analyses for comparison.²⁴”

4. For ref 27, “Macromolecules 28” should be moved after the title.

Response: We thank the reviewer very much for reading our manuscript very carefully. The reference was revised as follows:

“Ref 27) Schappacher, M. & Deffieux, A. Controlled synthesis of bicyclic "eight-shaped" poly(chloroethyl vinyl ether)s. *Macromolecules* **28**, 2629–2636 (1995).”

5. $M_{n,SEC}$ values of P4-a and MC4-a in Fig. 2 (9420 and 6710) are inconsistent with those in Fig. 3 (9360 and 6490), and D of MC4-a in Fig. 3 is 1.11, however, in Table 1, the value is 1.07.

Response: We apologize for the discrepancy in the values. We found that inconsistent values were determined because we had measured the SEC of the obtained polymers several times to verify the reproducibility. The $M_{n,SEC}$ and D values of **P4-a** and **MC4-a** in Fig. 3 are now corrected as follows:

For **P4-a**, “ $M_{n,SEC} = 9,420 \text{ g mol}^{-1}$ and $D = 1.05$ ”

For **MC4-a**, “ $M_{n,SEC} = 6,710 \text{ g mol}^{-1}$ and $D = 1.07$ ”

Additional changes are stated below.

1. Following sentences in the main text have been revised. (The page and line numbers reflect those in the revised manuscript)

1-1. Abstract

(line 5 in page 2) **Herein, we** establish a novel universal folding strategy~.

(lines 9–11 in page 2) This effective strategy marks a significant advancement **in** the development of functionalized materials **composed** of specific three-dimensional nanostructures.

(line 14 in page 2) **polymerization**

1-2. Introduction

(lines 2–4 in page 3) Precise folding of a biopolymer chain is an essential process to attain sophisticated higher-ordered structures, such as DNA packing and three-dimensional (3D) protein structures, which is responsible for their **outstanding** functions in living systems.¹⁻³

(lines 9–11 in page 3) However, the resulting SCNP is a statistical mixture of undefined-**shape** chains since the intramolecularly crosslinked formations randomly occur along the main chain.

(line 1 in page 5) the structure–property relationships~.

1-3. Results and Discussion

(lines 2–5 in page 6) The precursor of **P2** [**P2-a**; molecular weight from ¹H nuclear magnetic resonance (NMR) ($M_{n,NMR}$) = 6,200 g mol⁻¹, molecular weight from size-exclusion chromatography (SEC; $M_{n,SEC}$) = 9,970 g mol⁻¹, dispersity (D) = 1.06] was successfully synthesized by the ring-opening polymerization of ϵ -caprolactone~.

(line 9 in page 6) with slow addition of the polymer precursor (**Supplementary Scheme S5**).

(line 6 in page 7) hydrodynamically smaller polymer (i.e., **MC2-a**), ~.

(line 10 in page 7) matrix-assisted laser desorption/**ionization**-time of flight mass spectrometry~.

(line 16 in page 7) cyclic or tadpole-shaped PCLs (calculated $[M + Na]^+ = 5,174.11$, $n = 39$) formed **by** reaction~.

(line 19 in page 7) with the preferred isomer expected to **have** the tadpole topology~.

(line 3 in page 8) 8-shaped topology (**MC2-a**) was achieved **by** intramolecular ROMO

(line 7 in page 8) where the number of cyclic units are three and four, ~.

(line 2 in page 9) at the chain center and each terminus (**P3-a** and **P4-a**; $M_{n,NMR} = \sim 6,500$ g mol⁻¹) ~.

(line 9 in page 9) **Notably**, each MALDI-TOF MS spectrum~.

(line 16 in page 9) necessary, **which** is currently~.

(line 2 in page 10) Chain-end functionalization of topological polymers is essential to **facilitate** higher-order~.

(line 4 in page 10) groups **by using** functional Ru initiators~.

(lines 6–10 in page 11) By simply changing the degree of polymerization of the precursor, the molecular weight of each multicyclic (8-, trefoil-, and quatrefoil-shaped) polymer was successfully controlled from $\sim 6,000$ to 12,000 g mol⁻¹ (Table 1). Note that the suffix on the name of each polymer sample represents its molecular weight (**-a** for $\sim 6,000$ g mol⁻¹, **-b** for $\sim 9,000$ g mol⁻¹, and **-c** for $\sim 12,000$ g mol⁻¹).

(lines 4–5 in page 12) Here, it is important to note that the present folding strategy affords a polymer with a predetermined topology, as opposed to the synthesis of SCNPs, in which~.

(lines 7 in page 12) we performed intramolecular ROMO

(line 9 in page 12) used to determine~.

(line 12 in page 12) even in the case of *n*-hexane-rich media with up to~.

(line 3 in page 13) To get information~.

(line 7 in page 14) The *spiro*-multicyclic polymer is assumed to be a topological analog of~.

(line 1 in page 15) cyclic topology on polymer crystallization behavior (e.g., melting point, ~.

(line 7 in page 15) for the comparison.²⁴

(line 10 in page 15) crystallinity than those of cyclic PCLs (Fig. 4a, b).

(lines 10–11 in page 15) More specifically, MC2-a with molecular weight of ~6,000 g mol⁻¹ exhibited~.

(line 15 in page 15) resulting in lesser~.

(line 18 in page 15) Further increase in~.

(line 1 in page 17) intramolecular ROMO of the norbornenyl groups at predetermined positions.

(line 4 in page 17) on the structure–property relationships~.

(line 6 in page 17) Polymer folding into a *spiro*-type topology plays a crucial role in rendering higher-ordered functions to biomacromolecules.

2. The subheadings have been added in the Results and Discussion.
3. Sections titled “Methods” and “Data availability” have been added after the main text according to the guideline.
4. The section titled “Conflict of interest” has been replaced by “Competing Interests”.
5. A section titled “Author contribution” has been added after the main text to declare each author’s contribution in this work.
6. The author list in SI has been revised as follows:
Yoshinobu Mato,[†] Kohei Honda,[†] Brian J. Ree,[†] Kenji Tajima,[‡] Takuya Yamamoto,[‡] Tetsuo Deguchi,[§] Takuya Isono,^{‡,*} Toshifumi Satoh^{‡,*}
7. The indentation of synthetic details in SI has been unified.
8. Materials (S1-1) and Instruments (S1-1) in SI have been revised to add information about reagents and instruments used for the additional experiments.
9. In the section **Synthesis of HO-(PCL-OH)₂** in SI on page 10, the solvent amount has been revised.
10. In the section **Synthesis of I2** in SI on page 17, the unit of yield has been revised.

11. In the section **Synthesis of PMBO-(PCL-OH)₃** in SI on page 18, the unit of yield has been revised.
12. Following new references have been added:
- 33) Brown, H. A., De Crisci, A. G., Hedrick, J. L. & Waymouth, R. M. Amidine-mediated zwitterionic polymerization of lactide. *ACS Macro Lett.* **1**, 1113–1115 (2012).
 - 34) Misaka, H. et al. Synthesis of end-functionalized polyethers by phosphazene base-catalyzed ring-opening polymerization of 1,2-butylene oxide and glycidyl ether. *J. Polym. Sci. Part A Polym. Chem.* **50**, 1941–1952 (2012).
 - 38) Hoskins, N. J. & Grayson, M. S. Synthesis and degradation behavior of cyclic poly(ϵ -caprolactone). *Macromolecules* **42**, 6406–6413 (2009).
13. Following references have been revised.
- 5) Gonzalez-Burgos, M., Latorre-Sanchez, A. & Pomposo, J. A. Advances in single chain technology. *Chem. Soc. Rev.* **44**, 6122–6142 (2015).
 - 16) Ko, Y. S., Yamamoto, T. & Tezuka, Y. Click construction of spiro□ and bridged□ quatrefoil polymer topologies with kyklo□telechelics having an azide group. *Macromol. Rapid Commun.* **35**, 412–416 (2014).
 - 21) Kyoda, K., Yamamoto, T. & Tezuka, Y. Programmed polymer folding with periodically positioned tetrafunctional telechelic precursors by cyclic ammonium salt units as nodal points. *J. Am. Chem. Soc.* **141**, 7526–7536 (2019).
 - 25) Oike, H., Hamada, M., Eguchi, S., Danda, Y. & Tezuka, Y. Novel synthesis of single- and double-cyclic polystyrenes by electrostatic self-assembly and covalent fixation with telechelics having cyclic ammonium salt groups. *Macromolecules* **34**, 2776–2782 (2001).
 - 31) Ramakrishna, S. N., Morgese, G., Zenobi-Wong, M. & Benetti, E. M. Comblike polymers with topologically different side chains for surface modification: Assembly process and interfacial physicochemical properties. *Macromolecules* **52**, 1632–1641 (2019).
 - 32) Morgese, G., Cavalli, E., Rosenboom, J. G., Zenobi-Wong, M. & Benetti, E. M. Cyclic polymer grafts that lubricate and protect damaged cartilage. *Angew. Chem. Int. Ed.* **57**, 1621–1626 (2018).
14. The numbers of figures, schemes, tables, and references have been rearranged.
15. Table of Contents in SI has been revised.

REVIEWERS' COMMENTS:

Reviewer #1 (Remarks to the Author):

In this revised version, the authors have responded to the reviewers' comments and revised the manuscript adequately. I don't have any further comments or suggestion and I think this nicely prepared manuscript is now ready for publication.

Reviewer #2 (Remarks to the Author):

The authors have revised the manuscript according to my comments, I think it can be published in Communications Chemistry.